

# Long-term NOx measurements in the remote marine tropical troposphere

Simone T. Andersen[1*], Lucy J. Carpenter[1], Beth S. Nelson[1], Luis Neves[2], Katie A. Read[1,3], Chris Reed[4], Martyn Ward[1], Matthew J. Rowlinson[1,3], James D. Lee[1,3]

[1]Wolfson Atmospheric Chemistry Laboratories (WACL), Department of Chemistry, University of York, Heslington, York, YO10 5DD, UK.

[2] Instituto Nacional de Meteorologia e Geofísica, São Vicente (INMG), Mindelo, Cabo Verde.

[3]National Centre for Atmospheric Science (NCAS), University of York, Heslington, York, YO10 5DD, UK.

[4]FAAM Airborne Laboratory, Building 146, Cranfield University, Cranfield, MK43 0AL, UK.

[*]Corresponding author: sta516@york.ac.uk



## Abstract

Atmospheric nitrogen oxides (NO + NO₂ = NOₓ) have been measured at the Cape Verde Atmospheric Observatory (CVAO) in the tropical Atlantic (16° 51' N, 24° 52' W) since October 2006. These measurements represent a unique time series of $NO_x$ in the background remote troposphere. Nitrogen dioxide ($NO_2$) is measured via photolytic conversion to nitric oxide (NO) by ultra violet light emitting diode arrays followed by chemiluminescence detection. Since the measurements began, a "blue light converter" (BLC) has been used for $NO_2$ photolysis, with a maximum spectral output of 395 nm from 2006-2015 and of 385 nm from 2015. The original BLC used was constructed with a Teflon-like material and appeared to cause an overestimation of $NO_2$ when illuminated. To avoid such interferences, a new additional photolytic converter (PLC) with a quartz photolysis cell (maximum spectral output also 385 nm) was implemented in March 2017. Once corrections are made for the $NO_2$ artefact from the original BLC, the two $NO_2$ converters are shown to give comparable $NO_2$ mixing ratios (PLC = 0.92 × BLC, $R^2$ = 0.92), giving confidence in the quantitative measurement of $NO_x$ at very low levels. Data analysis methods for the $NO_x$ measurements made at CVAO have been developed and applied to the entire time series to produce an internally consistent and high quality long-term data set. NO has a clear diurnal pattern with a maximum mixing ratio of 2-10 pptV during the day depending on the season and ~0 pptV during the night. $NO_2$ shows a fairly flat diurnal signal, although a small increase in daytime $NO_x$ is evident in some months. Monthly average mixing ratios of $NO_2$ vary between 5 and 30 pptV depending on the season. Clear seasonal trends in NO and $NO_2$ levels can be observed with a maximum in autumn/winter and a minimum in spring/summer.




## 1  Introduction

Atmospheric nitrogen oxides play a key role in tropospheric chemistry. $NO_x$ helps to control the abundance of the two most important oxidants in the atmosphere, ozone ($O_3$) and the hydroxyl radical (OH). The presence of NO is usually the key limiting factor in the production of tropospheric $O_3$, which occurs via oxidation of NO to $NO_2$ by peroxy radicals ($RO_2$, $HO_2$) as described in reactions (1-2), followed by photolysis of $NO_2$ and rapid conversion of the resulting $O(^3P)$ to $O_3$:

$$RO_2 + NO \rightarrow RO + NO_2 \tag{1}$$

$$HO_2 + NO \rightarrow OH + NO_2 \tag{2}$$

$$NO_2 + hv \rightarrow NO + O(^3P) \qquad (hv \leq 410 \text{ nm}) \tag{3}$$

$$O(^3P) + O_2 + M \rightarrow O_3 + M \tag{4}$$

Reaction (2) also offers a route to the OH radical, above its primary production via $O_3$ photolysis (reactions (5) and (6)). If the $NO_x$ mixing ratio is sufficiently low, then peroxy radicals react with themselves instead of NO, and $O_3$ depleting reactions (reactions (5) to (8)) dominate over $O_3$ production (Atkinson, 2000).

$$O_3 + hv \rightarrow O(^1D) + O_2 \qquad (hv \leq 335 \text{ nm}) \tag{5}$$

$$O(^1D) + H_2O \rightarrow 2 \text{ OH} \tag{6}$$

$$OH + O_3 \rightarrow HO_2 + O_2 \tag{7}$$

$$HO_2 + O_3 \rightarrow OH + 2 O_2 \tag{8}$$

$NO_x$ mixing ratios below 10-30 pptV are generally sufficiently low for net tropospheric $O_3$ depletion (Atkinson, 2000; Jaeglé et al., 1998; Logan, 1985). These conditions have previously been reported to apply most of the year in the remote Atlantic Ocean (Lee et al., 2009). The mixing ratio of $NO_x$ in the atmosphere varies from a few pptV in remote areas (Lee et al., 2009; Monks et al., 1998; Reed et al., 2017) to >100 ppbV in polluted areas (Carslaw, 2005; Mazzeo et al., 2005; Pandey et al., 2008). It is therefore important to have representative $NO_x$ measurements in different regions of the world to be able to understand the chemistry occurring throughout the troposphere.

Long-term remote atmospheric $NO_x$ measurements are rare due to the difficulty measuring very low (pptV) mixing ratios. Many different methods of measuring $NO_x$ are available,





however, very few have the limit of detection (LOD) and sensitivity needed to measure $NO_x$
in remote regions. The most widely used method is NO chemiluminescence, where NO in the
presence of excess ozone is oxidized into excited state $NO_2$, which emits photons that can be
detected (Fontijn et al., 1970). $NO_2$ is generally converted into NO either catalytically by a
heated molybdenum converter or photolytically, followed by NO chemiluminescence (Kley
and McFarland, 1980). The molybdenum converter has historically been preferred due to its
high conversion efficiency of at least 95%, but it also converts other reactive nitrogen species
($NO_z$) such as peroxyacetyl nitrate (PAN), peroxymethacryloyl nitrate (MPAN), acyl peroxy
nitrates (APN), $HNO_3$, p-$HNO_3$, and HONO, potentially giving an overestimation of $NO_2$
(Dunlea et al., 2007; Grosjean and Harrison, 1985; Winer et al., 1974). Two separate studies
have shown that photolytic converters (PLC) with a wavelength of 385-395 nm have the
smallest spectral overlap with interfering compounds (Pollack et al., 2010; Reed et al., 2016).
Reed et al., (2016) showed that in some configurations the PLC can heat up the sampled air
making it possible for reactive nitrogen compounds such as peroxyacetyl nitrate (PAN) to
decompose thermally and cause an overestimation of $NO_2$. This, however, causes only a
negligible interference in warm regions such as Cabo Verde where PAN levels are extremely
low (Reed et al., 2016).
In this study we describe a $NO_2$ converter, similar to that presented by Pollack et al. (2010),
which has been implemented on an instrument to measure $NO_x$ at the CVAO. The data analysis
procedure is explained in detail and the first two years of results with the new converter are
presented and compared to the data obtained using a different converter.

## 89    2    Experimental

### 90    2.1    Location

The Cape Verde Atmospheric Observatory (CVAO; 16° 51' N, 24° 52' W) is located on
the north eastern coast of São Vicente, Cabo Verde. The air masses arriving at the CVAO
predominately come from the northeast (>95% of all wind direction measurements, see Figure
1) and have travelled over the Atlantic Ocean for multiple days since their last exposure to
anthropogenic emissions, with the potential exception of ship emissions (Carpenter et al., 2010;
Read et al., 2008). The UK Meteorological Office NAME dispersion model (Ryall et al., 2001)
has previously been used to investigate the origin of the air masses arriving at the CVAO,


which have been shown to be very diverse; North America, the Atlantic, Europe, Arctic, and
African regions (Lee et al., 2009). During the spring and summer, the air masses predominantly
originate from the Atlantic making it possible to investigate long-term remote marine
tropospheric background measurements. During the winter, the CVAO receives air mainly
from the Sahara, resulting in very high wintertime dust loadings (Chiapello et al., 1995; Fomba
et al., 2014; Rijkenberg et al., 2008). The time zone of Cabo Verde is always UTC-1. A full
description of the CVAO site and associated measurements is given in Carpenter et al. (2010).

## 2.2  Measurement Technique

NO$_x$ has been measured at the CVAO since 2006 using a NO$_x$ chemiluminescence
instrument manufactured by Air Quality Design Inc. (AQD), USA.  The chemiluminescence
technique involves the oxidation of NO by excess O$_3$ to excited NO$_2$ (Reaction 9) (Clough and
Thrush, 1967; Clyne et al., 1964; Fontijn et al., 1970). The excited NO$_2$ molecules can be
deactivated by emitting photons or by being quenched by other molecules (Reaction 10 and
11) such as N$_2$, O$_2$, and in particular H$_2$O. The emitted photons are detected by a
photomultiplier tube detector (PMT), which gives a signal linearly proportional to the mixing
ratio of NO sampled. The measurement of NO$_x$ and NO$_2$ requires photolytic conversion of NO$_2$
to NO (Reaction 3) followed by NO chemiluminescence detection (Kley and McFarland,

116   1980).

$$NO + O_3 \rightarrow NO_2^* + O_2 \tag{9}$$
$$NO_2^* \rightarrow NO_2 + h\nu \qquad (h\nu > 600 \text{ nm}) \tag{10}$$
$$NO_2^* + M \rightarrow NO_2 \tag{11}$$
$$NO_2 + h\nu \rightarrow NO + O(^3P) \qquad (h\nu \leq 410 \text{ nm}) \tag{3}$$
Further details of the technique are documented in (Carpenter et al., 2010; Drummond et al.,
1985; Fontijn et al., 1970; Lee et al., 2009; Peterson and Honrath, 1999; Reed et al., 2017; Val
Martin et al., 2006).





## 2.3 Instrument Set-up


Ambient air is sampled from a downward facing inlet placed into the prevailing wind with
a fitted hood 10 m above the ground. A centrifugal pump at a flow rate of ~750 litres per minute
pulls the air into a 40 mm glass manifold resulting in a linear sample flow of 10 m s$^{-1}$, giving
a residence time to the inlet of the $NO_x$ instrument of 2.3 s. To reduce the humidity and aerosol
concentration in the sampled air, dead-end traps are placed at the lowest point of the manifold
inside and outside the laboratory. A Nafion dryer (PD-50T-12-MKR, Permapure) is used to
additionally dry the sampled air, using a constant sheath flow of zero air (PAG 003, Eco Physics
AG) that has been filtered through a Sofnofil (Molecular Products) and activated charcoal
(Sigma Aldrich) trap (dewpoint -15°C). The air is sampled perpendicular to the manifold
through a 47mm PTFE (polytetrafluoroethylene) filter with a pore size of 1.2 μm.
A schematic diagram of the instrument is shown in Figure 2. Sampled air is passed through
two different photolytic $NO_2$ converters, which are placed in series. The first is a commercial
unit known as a BLC (Blue Light Converter) supplied by Air Quality Design, as described in
(Buhr, 2007). An ultra violet light emitting diode (UV-LED, 3 W, LED Engin, Inc.) array is
placed in each end of a reaction chamber made of Teflon-like barium doped material (BLC, λ
= 385 nm, volume = 16 cm$^3$). The entire block surrounding the reaction chamber is irradiated,
giving the highest possible conversion efficiency of $NO_2$. Each array is cooled by a heat sink
to maintain an approximately constant temperature inside of the converter when the diode
arrays turn on. The second converter consists of two diodes (Hamamatsu Lightningcure
L11921-500, λ = 385 nm) and a photolysis cell made of a quartz tube and two quartz windows
glued to each end with a volume of 16 cm$^3$ (PLC) following the design of Pollack et al. (2010).
Aluminium foil is wrapped around the quartz tube to increase the reflectivity to give the highest
conversion efficiency of $NO_2$. The diodes are placed at each end of the quartz tube, as shown
in Figure S2, without touching the windows to avoid increases in the temperature when the
diodes turn on. BLCs have been used at the CVAO since the instrument was installed in 2006,
with the most recent converter installed in April 2015 (a BLC2 model), where the wavelength
was changed to 385 nm from 395 nm. The PLC was installed in March 2017. The air flow
through the instrument is controlled at ~1000 sccm by a mass flow controller (MKS, M100B)
giving a residence time of 0.96 s through each of the converters.
To measure NO and $NO_x$ (NO + $NO_2$ converted into NO) the air is introduced to the
chemiluminescent detector (CLD), where NO is oxidized by excess $O_3$ into excited $NO_2$ in the
reaction volume (241 mL, aluminium with gold coating (Ridley and Grahek, 1990)) shown in
Figure 2. The reaction volume is kept at low pressure to minimize quenching of excited $NO_2$
and thereby maximize the NO chemiluminescence lifetime. The photons emitted from the
excited $NO_2$ molecules when they relax to ground state are detected by the PMT (Hamamatsu
R2257P) to give a signal for NO. $NO_2$ is converted into NO by the BLC for 1 minute, and then
the PLC for 1 minute, each period producing a signal due to NO + $NO_2$. The signal detected
by the PMT ($S_M$) is caused by NO reacting with $O_3$ ($S_{NO}$), dark current from the thermionic
emissions from the photocathode of the PMT ($S_D$), and an interference ($S_I$) which can be due
to other gas-phase reactions creating chemiluminescence and from illumination of the chamber
walls during $NO_2$ conversion (Drummond et al., 1985; Reed et al., 2016):
$$S_M = S_{NO} + S_D + S_I \qquad\qquad (I)$$
The PMT is cooled to -30°C to reduce the dark current, giving the instrument a higher
precision. Other molecules in the atmosphere such as alkenes also react with ozone and emit
photons to reach their ground state, but at a different time-scale to that of $NO_2$ (Alam et al.,
2020; Finlayson et al., 1974). This can give an interfering signal causing the NO and $NO_x$
mixing ratios to be overestimated. However, most of these reactions emit photons at 400-600
nm and are therefore filtered by a red transmission cut-off filter (Schott RG-610) placed in
front of the PMT (Alam et al., 2020). The filter transmits photons with a wavelength higher
than 600 nm (Drummond et al., 1985). A background measurement is therefore required to
account for the dark current of the PMT and for the remaining interfering reactions occurring
at a different time-scale to that of $NO_2$. Background measurements are made by allowing
ambient air to interact with $O_3$ in the zero volume (180 mL, PFA, Savillex, LLC) before
reaching the reaction volume (Figure 2). Most excited $NO_2$ molecules will reach their ground
state before the sample reaches the PMT, meaning the signal from NO will not be measured.
The efficiency of the reaction between NO and $O_3$ in the zero volume is calculated from the
calibration and will be explained in section 2.4.3.
NO, $NO_2$ and the background signal are all detected on the same channel, and the
instrument cycle is 1 min of background, 2 min of NO (when the $NO_2$ converters are off), 1
min of BLC $NO_x$ (the BLC converter is on), and 1 min of PLC $NO_x$ (the PLC is on).





## 2.4 Calibration


Prior to June 2019, calibrations were performed every 73 hours by standard addition in
order to account for temperature and humidity changes in the ambient matrix. In June 2019 the
calibration frequency was changed to every 61 hours to ensure that during any given month,
calibrations are carried out for approximately equal periods during the night and the day. To
calibrate the NO sensitivity, 8 sccm of 5 ppmV NO calibration gas in nitrogen is added to the
ambient air flow of ~1000 sccm, giving an NO mixing ratio of approximately 40 ppbV. The
mixing ratio used for calibrations are approximately 10,000 times that of the ambient
measurements, however, due to reduced cylinder stability for lower NO mixing ratios it is
difficult to calibrate at much lower mixing ratios and the chemiluminescence is expected to be
linear across the range of expected mixing ratios (Drummond et al., 1985). The calibration gas
is added between the PTFE filter and the $NO_2$ converter as shown in Figure 2. The conversion
efficiency of the BLC and the PLC is calibrated by gas phase titration (GPT), where oxygen is
added to the sampled NO calibration gas before entering the titration cell, which contains a UV
lamp that converts oxygen to ozone. Between 60-80% of the NO calibration gas is oxidized
into $NO_2$, giving a known mixing ratio of $NO_2$. A theoretical calibration sequence is shown in
Figure 3. The first cycle is to calibrate the sensitivity and the second is to calibrate the $NO_2$
conversion efficiency.  Each actual calibration includes three cycles of sensitivity calibration
and two cycles of conversion efficiency calibration. The signal from $NO_2$ observed in the NO
sensitivity calibration is due to traces of $NO_2$ in the calibration gas. Figure S3 shows the
observed percentage of $NO_2$ in the calibration cylinders from January 2014 to August 2019
calculated from the measured sensitivity (sec. 2.4.1) and the conversion efficiencies (CE) of
the two converters (sec. 2.4.2):
$$NO_2 \text{ in cylinder (pptV)} = \frac{(NO.c_{(1)} - NO_{(1)})}{\text{Sensitivity} \times CE} \qquad (II)$$
$$\text{Percentage } NO_2 = \frac{NO_2 \text{ in cylinder}}{NO_2 \text{ in cylinder} + NO \text{ cal conc.}} \qquad (III)$$
The percentage is stable for both converters, however, the PLC shows approximately 3-
4% $NO_2$ in the NO calibration gas compared to 5-10% for the BLC, which is caused by a BLC
artefact.  The cylinders used were certified to ≤2% $NO_2$.



### 2.4.1 Sensitivity


The sensitivity of the instrument is calculated from the increase in counts per second
caused by the calibration gas during NO calibration (untitrated, i.e. without $O_3$) and from the
mixing ratio of the calibration gas as shown by equation (IV). The NO counts per second from
the previous measurement cycle before the calibration is subtracted to give the increase due to
the calibration gas. The previous cycle needs to be stable and low in NO to give an accurate
sensitivity, which is the case at the CVAO.
$$\text{Sensitivity} = \frac{\text{Counts per second during calibration} - \text{Counts per second in previous cycle}}{\text{Mixing ratio of calibration gas}} \quad \text{(IV)}$$

The sensitivity of the instrument depends on the pressure of the reaction chamber, the
ozone mixing ratio in the reaction chamber, the flow of the sample through the reaction
chamber, and the temperature of the reaction chamber. To maintain a stable sensitivity, all four
parameters should be kept stable (Galbally, 2019). From January 2014 to August 2019 the
sensitivity has varied between 2.7 and 7.4 counts $s^{-1}$ $pptV^{-1}$ with changes of less than 5%
between subsequent calibrations (see Figure S4), unless the instrument has been turned off for
a long period of time due to instrumental problems.

### 2.4.2 Conversion Efficiencies


The conversion efficiency of the BLC and the PLC is calculated based on the titrated (with
added $O_3$) and the untitrated (without added $O_3$) NO calibration gas as described in equation
(V). The numerator gives how much of the NO is titrated into $NO_2$ and the denominator
represents the $NO_2$ measured when taking the $NO_2$ content in the NO calibration gas into
account. In equation (V), "NO" is the measurement of only NO i.e. when the converters are
off, NO.c is when one of the converters are on therefore the measurement is $NO + NO_2$ and (1)
and (2) represent untitrated and titrated NO, respectively.
$$CE = \frac{[(\text{NO.c}_{(2)} - \text{NO}_{(2)}) - (\text{NO.c}_{(1)} - \text{NO}_{(1)})]}{[\text{NO}_{(1)} - \text{NO}_{(2)}]} = 1 - \frac{\text{NO.c}_{(1)} - \text{NO.c}_{(2)}}{\text{NO}_{(1)} - \text{NO}_{(2)}} \quad \text{(V)}$$

The conversion efficiency of the BLC has varied from 82% to 91% between its installation in
April 2015 and August 2019 ($j$ ~3 $s^{-1}$). Prior to April 2015, an older generation BLC ($\lambda = 395$
nm) with a conversion efficiency of 30-35% was used ($j$ ~0.5 $s^{-1}$). The conversion efficiency
of the PLC has varied between 50% and 55% from its installation in March 2017 to August
2019 ($j$ ~1 $s^{-1}$). See Figure S5 for all the calculated conversion efficiencies.




### 2.4.3 Efficiency of the Zero Volume

Background measurements are made by reacting NO and interference compounds with $O_3$
in the zero volume (Figure 2). The system is set up so that $NO_2$ produced from NO will relax
to the ground state before it is measured in the downstream reaction chamber, whereas it is
assumed that any interfering compounds will emit photons when reaching the reaction chamber
and be measured as a background signal (Drummond et al., 1985; Galbally, 2019). If the zero
volume is too small or the $O_3$ mixing ratio is too low, some untitrated NO may lead to $NO_2$
chemiluminescence within the reaction chamber and the background will be overestimated. On
the other hand, if the zero volume is too large, some of the interfering compounds may have
relaxed to their ground state before the reaction chamber and the background signal will be
underestimated. The residence time of zero volume is 10.8s compared to 14.5s for the reaction
volume. The efficiency of the zero volume can be calculated from the calibration cycle. The
difference in background counts from before a calibration cycle to during the calibration cycle
shows how much of the added NO from the calibration cylinder does not react with $O_3$ in the
zero volume. By dividing this difference by the signal due to NO during the NO measurement
of the calibration cycle, which is obtained by subtracting the NO measurement from the
previous measurement cycle, the inefficiency of the zero volume is obtained. The efficiency is
determined for each calibration cycle (eq. VI) and plotted in Figure S6. It is consistently above

265 98%.

$$\text{Efficiency}_{ZV} = 1 - \frac{\text{cal zero} - \text{measurement zero}}{\text{NO cal} - \text{previous NO cycle}} \qquad \text{(VI)}$$

### 2.4.4 Artefact Measurements

As described in section 2.3, $NO_x$ measurements may have artefacts from
chemiluminescence caused by interfering gas-phase reactions and/or from compounds
produced by illumination of the reaction chamber walls as well as from pressure differences in
the instrument (Drummond et al., 1985; Reed et al., 2016). To estimate artefacts, it is necessary
to measure the signal from $NO_x$-free air. The calibration sequence is followed by sampling
$NO_x$-free air generated from a pure air generator (PAG 003, Eco Physics AG) for 30 minutes.
According to the manufacturer, the PAG not only scrubs NO, $NO_2$ and $NO_y$ from the ambient
air but also $SO_2$, VOCs, $H_2O$ and $O_3$. An overflow of PAG air is introduced between the aerosol





filter and the $NO_2$ converters as shown in Figure 2 and the cycle of background, NO, $NO_x$ BLC, and $NO_x$ PLC is used to estimate artefact NO and $NO_2$ measured by the instrument.

### 2.4.4.1    NO Artefact

The NO artefact can be caused by two things; alkenes reacting with $O_3$ and giving chemiluminescence above 600 nm at approximately the same rate as $NO_2$ or a difference in pressure between the zero volume and the reaction volume. An artefact caused by alkenes will be positive and overestimate the NO mixing ratio, where an artefact due to a pressure difference can be either negative or positive. It can be estimated as the offset from 0 pptV when the mixing ratio sampled is 0 pptV. The NO mixing ratio is expected to be 0 pptV when sampling $NO_x$-free air or between 22.00 and 04.00 UTC at night. NO generated during the day is rapidly oxidized into $NO_2$ through reactions with $O_3$ and $RO_2$ after sunset. During the night, NO is not generated from photolysis of $NO_2$, and there are no significant local sources of NO at Cabo Verde when the air masses come from over the ocean (which is >95% of the time). The average NO mixing ratio between 22.00 and 04.00 UTC and the average NO mixing ratio from the PAG zero air tend to be very similar, with the PAG artefact (-3.68 ± 22.91 pptV ($2\sigma$), January 2014 – August 2019) ordinarily lower than the night time artefact (0.39 ± 11.92 pptV ($2\sigma$), January 2014 – August 2019). Time series of both NO artefact measurements can be found in Figure S7 in the supplementary information. The night time NO artefact is used as it is measured more frequently, it contains the same ambient matrix with nothing scrubbed and to eliminate the possibility of residual NO influencing background measurements determined from the PAG. Since the PAG scrubs VOCs it will also not give an estimate of the artefacts caused by fast reacting alkenes.

### 2.4.4.2    $NO_2$ Artefact

$NO_2$ converters have previously been shown to have artefacts caused by thermal or photolytic conversion of reactive nitrogen compounds ($NO_z$) other than $NO_2$ as well as illumination of the chamber walls (Drummond et al., 1985; Reed et al., 2016; Ryerson et al., 2000). Fast reacting alkenes, which can cause overestimations of the NO mixing ratios, will not cause the $NO_2$ mixing ratio to be overestimated, since the NO signal is subtracted from the $NO_2$ signal.

The spectral output of an $NO_2$ converter with a wavelength of 385 nm was compared to
absorption cross sections of $NO_2$ and potential interfering species such as $BrONO_2$, HONO and
$NO_3$ (Reed et al., 2016). The photolytic convertor was shown to have good spectral overlap
with the $NO_2$ cross section with minimal spectral overlap with other $NO_z$ species, except for a
small overlap with the absorption cross section of HONO. The interference from $BrONO_2$,
HONO and $NO_3$ have additionally been evaluated previously for a similar set-up using a Hg
lamp (Ryerson et al., 2000). At equal concentrations of $NO_2$ and $NO_z$ species, $BrONO_2$ and
$NO_3$ were estimated to maximum have an interference of 5% and 10%, respectively, using a
lamp with a wider spectral overlap with the interfering species than what is observed for the
LEDs used at the CVAO (Ryerson et al., 2000). At the CVAO, HONO levels have previously
been measured to peak at ~3.5 pptV (at noon; (Reed et al., 2017)). For the typical Gaussian
output of a UV-LED this interference is calculated to be 2.0, 12.6, and 25.7% for UV-LEDs
with principle outputs of 395, 385, and 365 nm respectively, resulting in a maximum
interference of <0.5 pptV during peak daylight hours. Photolytic conversion of $NO_z$ species is
therefore not expected to be an important contributor to the $NO_2$ artefact at the CVAO due to
the narrow spectral output of the LEDs.
Each converter is only on for 1 minute in a 5-minute cycle. For thermal conversion to be
a major contributor to the artefact, the converter would have to increase in temperature during
that one minute and not the rest of the cycle otherwise an increase in signal should be constant
since the air continues to flow through the converters when they are turned off. Thermal
decomposition of $NO_z$ species is therefore not expected to have an effect in a climate like the
one in Cabo Verde, where the sample temperatures are similar to the ambient temperatures.
It has been shown that the walls of a BLC made out of a porous Teflon-like doped block
becomes contaminated from the ambient air over time and when the walls are illuminated
reactions take place on the surface causing an artefact (Reed et al., 2016; Ryerson et al., 2000).
The BLC is similar to the one used by Reed et al. (2016) and it is therefore expected to have
an artefact due to reactions taking place on the surface. The PLC is not expected to be
contaminated in the same way as it does not have porous chamber walls. Ryerson et al. (2000)
observed an increase in artefact over time when sampling ambient air for a similar PLC,
however, this is not observed for the PLC in the very clean environment at the CVAO (0-10
pptV between August 2017 and August 2019, see below) and surface reactions are therefore
expected to give a negligible artefact for the PLC.



The total artefact can be determined by measuring the $NO_2$ signal when the $NO_2$ mixing
ratio is 0 pptV, however, it is virtually impossible to scrub all $NO_x$ from the ambient air and
nothing else. To estimate the $NO_2$ artefact, PAG zero air is measured using both converters.
The PLC measures between 0-10 pptV compared to 10-60 pptV using the BLC. Since, as
discussed above, the $NO_2$ artefact of the PLC is believed to be negligible, the signal is believed
to represent the remaining $NO_2$ in the zero air after scrubbing. The signal from the BLC when
measuring PAG zero air is expected to be due to the illumination of the chamber walls in
addition to the traces of $NO_2$ left in the zero air. The artefact due to wall reactions in the BLC
can therefore be estimated by subtracting the signal measured by the PLC.

## 3   Data Analysis
Time periods with known problems such as maintenance on the manifold, ozone leaks,
and periods when the PMT has not reached <-28°C are not included in the dataset. The mean
and standard deviation of the zero (background), NO, $NO_2$ BLC and PLC are determined for
each 5-minute measurement cycle.  To avoid averaging over the time it takes the detector to
change and stabilize between the different types of measurements, the last 50 seconds of the
measurement cycle are used for the background and the NO counts, and the last 30 seconds for
the BLC $NO_x$ and the PLC $NO_x$ counts.  Each cycle is filtered based on the percentage standard
deviations and differences in counts between subsequent cycles. If the standard deviation or
the difference in counts are outside the mean $\pm$ 2σ (see Table 1) calculated from a 5-year period
between 2014 and 2018, the cycle is not used for further analysis. This removes noisy data as
well as sharp spikes but keeps data with sustained increases lasting more than 5 minutes.
To obtain the signals due to NO and $NO_2$, the interpolated zero and NO measurements are
subtracted from the NO and $NO_x$ measurements, respectively. They are converted to a
concentration by using the interpolated sensitivity and conversion efficiency as shown in
equation VII and VIII:
$$NO \text{ mixing ratio} = \frac{NO \text{ measurement} - \text{background measurement}}{\text{Sensitivity}} \qquad \text{(VII)}$$
$$NO_2 \text{ Mixing ratio} = \frac{NO_x \text{ measurement} - NO \text{ measurement}}{\text{Sensitivity} \times CE} \qquad \text{(VIII)}$$
The NO and $NO_2$ BLC concentrations are corrected by subtracting the interpolated
artefacts described in sections 2.4.4.1 and 2.4.4.2. If the difference between two subsequent
NO artefact measurements vary by more than the mean $\pm 2\sigma$ of the differences in NO artefacts
determined from January 2014 – August 2019 ($0.00 \pm 6.18$ pptV), the measurements made
between will not be used for further analysis due to a potential step change between the
determinations.

374         Hourly averages of all the measurements are determined. If data coverage during the hour

is less than 50%, the hour is flagged and discarded from the data analysis. The hourly $NO_x$ (NO
+ $NO_2$ PLC) concentrations between June 2017 and August 2019 are plotted as a function of
wind speed and direction in figure 4. It can be observed that the concentrations are enhanced
at low wind speed and when the air crosses the island (southwest). Measurements made at a
wind speed <2 m/s or from a wind direction >100° are, therefore, flagged as suspected of local
contamination and are not used in the analysis. Extreme mixing ratios outside the mean $\pm 4\sigma$
of the 5-year for NO and 2-year period for $NO_2$ are flagged as suspicious (see Table 1 for
boundaries). Lastly, inconsistence in the measurements such as differences outside the mean $\pm$
$4\sigma$ between the mean and median of a measurement (see Table 1 for boundaries) and
differences between the two $NO_2$ measurements are flagged as suspicious ($0.4 \pm 32.2$ pptV).
The data remaining after these removals are 88% of the original NO and $NO_2$ BLC dataset and
83% of the $NO_2$ PLC dataset remain to analyse.

## 3.1 Corrections

389         As described above, excited $NO_2$ can be quenched by other sampled molecules, giving a

lower observed mixing ratio than the real value. Water molecules are effective quenchers and
therefore a correction is usually applied depending on the humidity (Matthews et al., 1977;
Ridley et al., 1992). However, since the calibrations at the CVAO are performed by standard
addition, and a Nafion dryer is placed in front of the instrument, this is not necessary.

394         Additionally, NO can react with $O_3$ in the ambient air in the inlet and manifold giving an

overestimation of $NO_2$ and an underestimation of NO. To correct for this the following
equations are used (Gilge et al., 2014):
$$[NO]_0 = [NO]_{E1} \times e^{k_{O_3} \times t_{E1}} \tag{IX}$$
$$[NO_2]_0 = \left(\frac{J_C + k_{O_3}}{J_C}\right) \times \left(\frac{[NO]_{E2} - [NO]_{E1} \times e^{-\left(k_{O_3} \times (t_{C2} - t_{C1}) + J_C \times t_{C2}\right)}}{1 - e^{\left(-(k_{O_3} + J_C) \times t_{C2}\right)}}\right) - [NO]_0 \tag{X}$$





where $[NO]_0$ is the corrected NO mixing ratio, $[NO]_{E1}$ is the uncorrected NO mixing ratio,
$[NO_2]_0$ is the corrected $NO_2$ mixing ratio, $[NO]_{E2}$ is the uncorrected NO mixing ratio when the
converter is on, $k_{O3}$ is the rate of the reaction between NO and $O_3$ ($k(O_3+NO) \times [O_3] \times 10^{-9} \times$
M), $t_{E1}$ is the sum of the residence time from the inlet to entry of the converter and the time the
air is in the converter, $t_{C1}$ and $t_{C2}$ are the time the air is in the converter when the converter is
on and off, respectively, and $J_C$ is the photolysis rate inside the converter. The residence time
from the inlet to the entry of the converter has been 2.3 s since 2015 and the time the air is in
each of the converters is 1.0 s (with and without the converter on). The $O_3$ mixing ratio
measured at the CVAO has varied between 5 and 60 ppbV (with an uncertainty of 0.07 ppbV)
between 2014 and 2019. The ozone correction is calculated for each hour using a rate
coefficient of $1.8 \times 10^{-14}$ cm$^3$ molecule$^{-1}$ s$^{-1}$ at 298K (Atkinson et al., 2004). This gives an
average $O_3$ correction $\pm 2\sigma$ of $6.8 \pm 3.0\%$, $1.7 \pm 11.0\%$, and $1.3 \pm 7.1\%$ for NO, $NO_2$ BLC, and
$NO_2$ PLC, respectively, when the mixing ratio measured is above 0.1 pptV (See supplementary
information for an example of the calculation). Thus, at the low mixing ratios of $O_3$ present at
Cabo Verde and the short residence time for sampling, the corrections for $O_3$ are well within
the noise of the measurements (see below), but are still included in the final calculated mixing
ratios.

## 4   Uncertainty Analysis

To be able to evaluate the $NO_x$ measurements made at the CVAO an extensive uncertainty
analysis is performed. A summary of the analysis can be found in Table 2 and a detailed
description in the supplementary information. The hourly precision and uncertainty of the
instrument are estimated to characterize the uncertainties at the 95 percent confidence interval
(Bell, 1999). The hourly precision is estimated from the zero count variability, which is directly
related to the photon-counting precision of the PMT. The uncertainty of the hourly
measurements is estimated by combining all the uncertainties associated with the
measurements. This includes uncertainties in the calibrations, artefact determinations, and $O_3$
corrections as well as the precision of the instrument. The precision of the NO and $NO_2$
measurements are both included in the total uncertainty of the $NO_2$ measurements as the NO
measurements are subtracted from the $NO_2$ measurements. Each term is converted into pptV
to be able to combine them using error propagation.



The 2σ precision for hourly averaged NO data between January 2014 and August 2019 is
0.96 ± 0.89 pptV. The hourly precisions reported here are in good agreement with our
previously reported 1σ precision of the instrument of 0.30 pptV (Reed et al., 2017) and the 2σ
precision of 0.6-1.7 pptV (Lee et al., 2009). The $NO_2$ precisions are determined by taking the
conversion efficiency of the respective converters into account. The hourly 2σ precision for
hourly averaged $NO_2$ data between March 2017 and August 2019 becomes 1.45 ± 0.82 pptV
and 2.74 ± 2.18 pptV for the BLC and PLC, respectively. The determined $NO_2$ precisions are
within the interval of previously reported precisions for the same instrument (Lee et al., 2009;
Reed et al., 2017).
The total hourly uncertainty for each of the three measurements are determined by
combining all the uncertainties described using propagation of uncertainties. The precisions
are already calculated as hourly precisions in pptV. The calibration uncertainties are
interpolated between each calibration and multiplied by the hourly concentrations of NO and
$NO_2$ to calculate hourly uncertainties in pptV. The artefact uncertainties are interpolated
between each artefact determination, and the uncertainty due to ozone corrections is determined
by multiplying the % uncertainties by the hourly concentrations of NO and $NO_2$. The hourly
uncertainties are determined to be 1.42 ± 1.47 pptV, 8.38 ± 7.46 pptV, and 4.44 ± 5.79 pptV
for NO, $NO_2$ BLC, and $NO_2$ PLC, respectively.

## 5    Results: Examples of Data

The first year of data (August $1^{st}$ 2017 to July $31^{st}$ 2018) is chosen as an example of the
resulting NO and $NO_2$ datasets. October 2017, December 2017, and April 2018 are used to
highlight the seasonality in the mixing ratios observed during a year of measurements. Panel
A in figure 5 and 6 show the full $O_3$ corrected time series for NO and $NO_2$, respectively. Panel
B, C, and D in the two figures show the time series for the three chosen months and panel E,
F, and G show the 3-hour rolling average diurnals for the same months. Monthly diurnals for
NO and $NO_2$ for the entire year can be found in figure S8 and S9, respectively.
Clear seasonality can be observed in the diurnal cycles of NO measurements with a
maximum of ~10 pptV in Winter and a minimum of ~2 pptV in the spring and summer. This
is in good agreement with that reported for previous years (Lee et al., 2009; Reed et al., 2017).
The two $NO_2$ measurements are in general in good agreement when looking at the time series

461 in figure 6. Offsets of up to 10 pptV between the two measurements can be seen over some

462 time periods (E.g. April, Panel D), which are most likely caused by the calculated BLC artefact

463 for those periods either being too high or too low. This is supported by the diurnals having the

464 same shape, but with an offset. Monthly diurnals of the two $NO_2$ measurements agree within 2

465 standard errors except in August 2017, where the offset between the two measurements is larger

466 than for the remaining months. $NO_2$ shows a fairly flat diurnal signal, although a small increase

467 in daytime $NO_2$ is evident in some months, which is in agreement with that reported for

468 previous years (Lee et al., 2009; Reed et al., 2017). Spikes in the early morning are noticeable

469 in the $NO_2$ diurnals for July-November, which correspond to the months with an average lower

470 wind speed than the rest of the year (the diurnal for April also shows a spike, however, it is

471 caused by only one morning). These spikes could be caused by local fishing boats passing

472 upwind of the observatory in the morning hours, which will give a more prominent spike a low

473 wind speed. Monthly wind speed diurnals can be found in figure S10. The good agreement

474 between the two $NO_2$ measurements observed in figure 6 can also be observed in figure 7,

475 where the two are plotted against each other. The data points are scattered around the 1:1 line

476 shown in black with an overestimation by the BLC. An orthogonal distance regression (ODR)

477 is performed to evaluate the scatter of the data points with uncertainty in both measurements

478 between August 2017 and 2019. The resulting regression line is displayed in red (PLC $NO_2$ =

479 $0.92 \times$ BLC $NO_2$, $R^2 = 0.92$).

480  The seasonality of the NO measurements can be explained by a combination of the

481 variation of the origin of the air masses arriving at the CVAO, meteorology, photolysis rates,

482 and seasonality of emissions. Back trajectories of the three months used as examples are shown

483 in figure 8. FLEXPART version 10.4 is used in backwards mode, driven by pressure level data

484 from Global Forecast System (GFS) reanalyses at 0.5°×0.5° resolution (Pisso et al., 2019; Stohl

485 et al., 1998). 10-day back-trajectory simulations are initialised every 6 hours, releasing 1000

486 particles from the CVAO site. Further information on FLEXPART can be found in the

487 supplementary information. During the winter maximum (December) the back-trajectories

488 indicate that the air reaching CVAO is largely dominated by African air, compared to during

489 the spring minimum (April), which is dominated by Atlantic marine air. Large west African

490 cities such as Dakar and Nouakchott, and/or the shipping lanes to the east/northeast of Cabo

491 Verde, are potential candidates for the source of elevated $NO_x$. The NO mixing ratios measured

492 in October are higher than those in April and lower than in December. This may be due in part

493 to the influence of polluted African air arriving at Cabo Verde, which is more prominent in





October than in April, but less so than in December. The $NO_2$ and the total $NO_x$ (NO + PLC
$NO_2$, figure 9) similarly show higher levels in December than April, but the mixing ratios
observed in October are similar to those in April. It should be noted that some of the days with
high percentages of African air have missing data or wind directions from other places than the
north east.
From table 3 it can be observed that the NO, $NO_2$, and $NO_x$ measurements at the CVAO
compare well to the few other measurements in the remote marine boundary layer as well as
background sites in Alert, Canada and measurements in the free troposphere. A wintertime
seasonal increase in NO, $NO_2$, and $NO_x$ can be observed during December-February, which
corresponds to the months when surface air masses arrive at Cabo Verde from western Africa
(Carpenter et al., 2010; Lee et al., 2009).

## 6  Conclusion


A photolytic $NO_2$ converter with external diodes and a quartz photolysis cell (PLC) has
been installed at the Cape Verde Atmospheric Observatory and the $NO_2$ measurements have
been compared to those of the historical BLC used at the site, which has internal diodes and a
reaction chamber made of Teflon-like barium doped material. The two measurements show
good agreement (PLC $NO_2$ = 0.92 × BLC $NO_2$, $R^2$ = 0.92) with small differences due to
uncertainties in the estimations of the BLC $NO_2$ artefact. Even though the PLC has a lower
conversion efficiency (CE= 52 ± 4%) than the BLC (CE= 85 ± 4%), it is preferred due to its
assumed negligible artefact as a consequence of having non-porous/non-reactive walls. The
assumption of a zero artefact causes the hourly uncertainty of the $NO_2$ measurements to be
roughly halved. With $2\sigma$ hourly precisions of 0.96 ± 0.89 pptV, 1.45 ± 0.82 pptV, and 2.74 ±
2.18 pptV and $2\sigma$ hourly uncertainties of 1.42 ± 1.47 pptV, 8.38 ± 7.46 pptV, and 4.44 ± 5.79
pptV for NO, $NO_2$ BLC, and $NO_2$ PLC, respectively, the instrument has a high repeatability
and low uncertainties for all the measurements. The mixing ratios observed at the CVAO (NO:
2-10 pptV, $NO_2$: 5-50 pptV, and $NO_x$: 7-60 pptV at midday) are in agreement with previous
measurements at the CVAO as well as other previous remote measurements around the world.





## 7 Data availability

The processed data is available through Ebas
(http://ebas.nilu.no/Pages/DataSetList.aspx?key=45DB99FE2B7F4F97864ECF800E71E5D5
) and through CEDA (Center for Environmental Data Analysis,
https://catalogue.ceda.ac.uk/uuid/d5422d54d519ed056cc17e97037732b8).

## 8 Author contribution

LN runs the instrument on a day-to-day basis. MW and STA wrote the script processing the data. MJR ran the back trajectory analysis. BSN developed the photolytic converter setup. All authors were involved in the analysis, data evaluation and discussion of the results. STA wrote the paper with contributions from all coauthors. All coauthors proofread and commented on the paper.

## 9 Competing interests

The authors declare that they have no conflict of interest.

## 10 Acknowledgements

The authors would like to thank Franz Rohrer (Forschungzentrum Jülich) and Tomás Sherwen (University of York) for scientific discussions, and NERC/NCAS for funding the CVAO programme. STA's PhD was funded through the NERC SPHERES Doctoral Training Partnership and the University of York.





## 11 Tables

Table 1: The mean ± 2σ of the standard deviation and difference in counts/s between two subsequent measurement cycles.

| Measurement | Standard deviation (%)[a] | Difference in counts/s[b] | Extreme values (pptV)[c] | Extreme difference between mean and median (pptV)[d] |
|---|---|---|---|---|
| Zero | 2.4 ± 1.7 | - | - | - |
| NO | 2.5 ± 10.6 | 0 ± 515 | 1.7 ± 47.9 | 0.2 ± 4.1 |
| NO$_2$ BLC | 2.5 ± 7.5 | 0 ± 1432 | 16.8 ± 175.2 | 1.5 ± 33.0 |
| NO$_2$ PLC | 2.1 ± 2.5 | 0 ± 738 | 17.3 ± 176.8 | 1.7 ± 33.0 |

[a]Determined as the standard deviation of a cycle divided by the mean. [b]The difference in counts/s between each cycle. [c]Calculated as the hourly mean ± 4 standard deviations of the hourly mixing ratio. [d]Calculated as the hourly mean ± 4 standard deviations of the differences between the mean and median.





Table 2: Calculated uncertainties associated with the $NO_x$ measurements. The values in bold are the combined uncertainties for each type of measurement. Each uncertainty is given as the mean uncertainty $\pm$ 2 standard deviation of the data between January 2014 and August 2019 for NO and from March 2017 to August 2019 for both $NO_2$ measurements.

| Source of uncertainty | Probability distribution | Uncertainty (%) | Uncertainty (pptV) |
|---|---|---|---|
| Hourly precision/repeatability NO | Normal | | 0.96 ± 0.89 |
| Hourly precision/repeatability $NO_2$ BLC | Normal | | 1.45 ± 0.82 |
| Hourly precision/repeatability $NO_2$ PLC | Normal | | 2.74 ± 2.18 |
| Total Calibration uncertainty NO[a] | | 2.78 ± 8.05 | 0.03 ± 0.27 |
| Total Calibration uncertainty $NO_2$ BLC[a] | | 3.44 ± 9.32 | 0.33 ± 1.27 |
| Total Calibration uncertainty $NO_2$ PLC[a] | | 3.52 ± 8.67 | 0.37 ± 1.27 |
| Total NO artefact uncertainty[b] | | | 1.05 ± 3.44 |
| Total $NO_2$ artefact uncertainty[b] | | | 7.19 ± 7.24 |
| Hourly $O_3$ correction uncertainty NO | Normal | 20.00 ± 0.001 | 0.27 ± 1.14 |
| Hourly $O_3$ correction uncertainty $NO_2$ BLC | Normal | 20.00 ± 0.001 | 2.47 ± 6.75 |
| Hourly $O_3$ correction uncertainty $NO_2$ PLC | Normal | 20.00 ± 0.001 | 2.60 ± 6.37 |
| **Total hourly uncertainty NO** | | | **1.42 ± 1.47** |
| **Total hourly uncertainty $NO_2$ BLC** | | | **8.38 ± 7.46** |
| **Total hourly uncertainty $NO_2$ PLC** | | | **4.44 ± 5.79** |

[a]The individual uncertainties associated with the calibration can be found in table S1. [b]The individual uncertainties associated with the artefact determination can be found in table S2.



Table 3: NO, $NO_2$, and $NO_x$ mixing ratios at different low $NO_x$ sites.

| | NO (pptV)[k] | NO$_2$ (pptV) | NO$_x$ (pptV) | Reference |
|---|---|---|---|---|
| **Tropospheric Marine** | | | | |
| CVAO, Cape Verde 2017-2018 | 2-10 | 5-50 | 7-60 | This study |
| Cape Grim, Australia[a] | 1-6 | 3-6 | 4-12 | (Monks et al., 1998) |
| SAGA3, Pacific Ocean, Cruise[b] | 2.9 ± 0.1 | | | (Torres and Thompson, 1993) |
| ASTEX, North Atlantic, Cruise[c] | 5 ± 4 | 29 ± 8 | | (Carsey et al., 1997) |
| WOCE, Indian Ocean, Cruise[d] | ~ 5 | 18-40 | | (Rhoads et al., 1997) |
| **Background Sites** | | | | |
| Alert, Canada[e] | 0.2-2.8 | 1.3-10.8 | | (Beine et al., 2002) |
| South Pole[f] | ~ 10 | | | (Jones et al., 1999) |
| **Free Troposphere** | | | | |
| Mauna Loa, USA[g] | 9.4 | 29.6 | 32 | (Carroll et al., 1992) |
| Pico Mountain, Portugal[h] | 0-9 | 19-30 | 20-37 | (Val Martin et al., 2008) |
| NASA GTE, Pacific Ocean, Aircraft[i] | ~ 1 | | | (Ridley et al., 1987) |
| Svalbard, Norway[j] | | | 27.7 ± 24.0 | (Beine et al., 1996) |

[a]Measurements made during the SOAPEX (Southern Ocean Atmospheric Photochemistry EXperiment) campaign during Austral summer in 1995. [b]Measurements from the Soviet-American Gases and Aerosols (SAGA) campaign between Hawaii and American Samoa between February and March. [c]Measurements from 6 clean days on the Atlantic Stratocumulus Transition Experiment (ASTEX). [d]Measurements from the World Ocean Circulation Experiment (WOCE) between South Africa and Sri Lanka. [e]Measurements made during 24-hour darkness and in spring. [f]Measurements made from January-March 1997 at the German Antarctic research station, Neumayer. [g]Measurements made during the Mauna Loa Observatory Photochemistry Experiment (MLOPEX) in May 1988. [h]Measurements made at Mount Pico between July 2002 and August 2005. [i]Measurements made in the upper marine boundary layer from 13 flights between California and west of Hawaii. [j]Measurements made at the Ny-Ålesund Zeppelin mountain station on Svalbard during a spring campaign in 1994. [k]Daytime values.



# 12 Figures

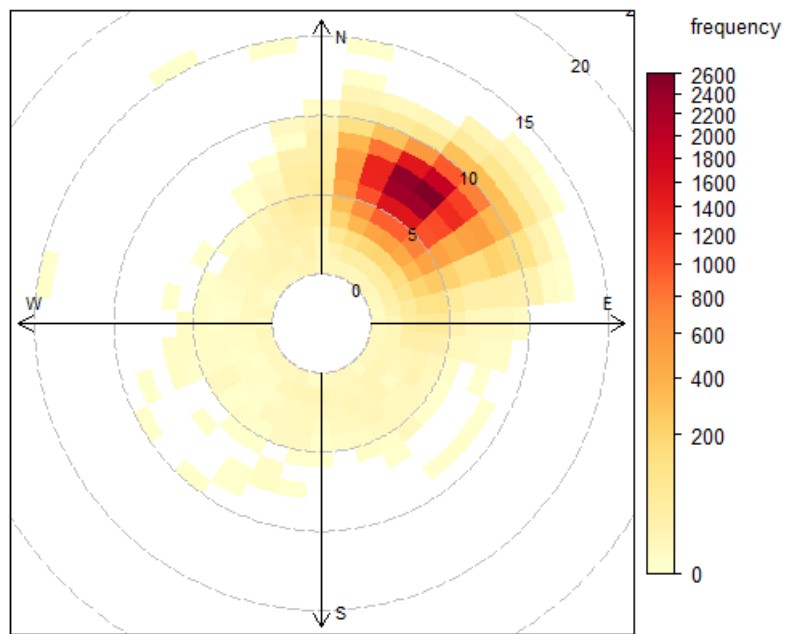

Figure 1: The frequency of hourly averaged wind speed and direction from January 2014 to August 2019. Each square symbolise 10 degrees of wind direction and 1 m/s wind speed. Each dashed circle show an increase in wind speed of 5 m/s.



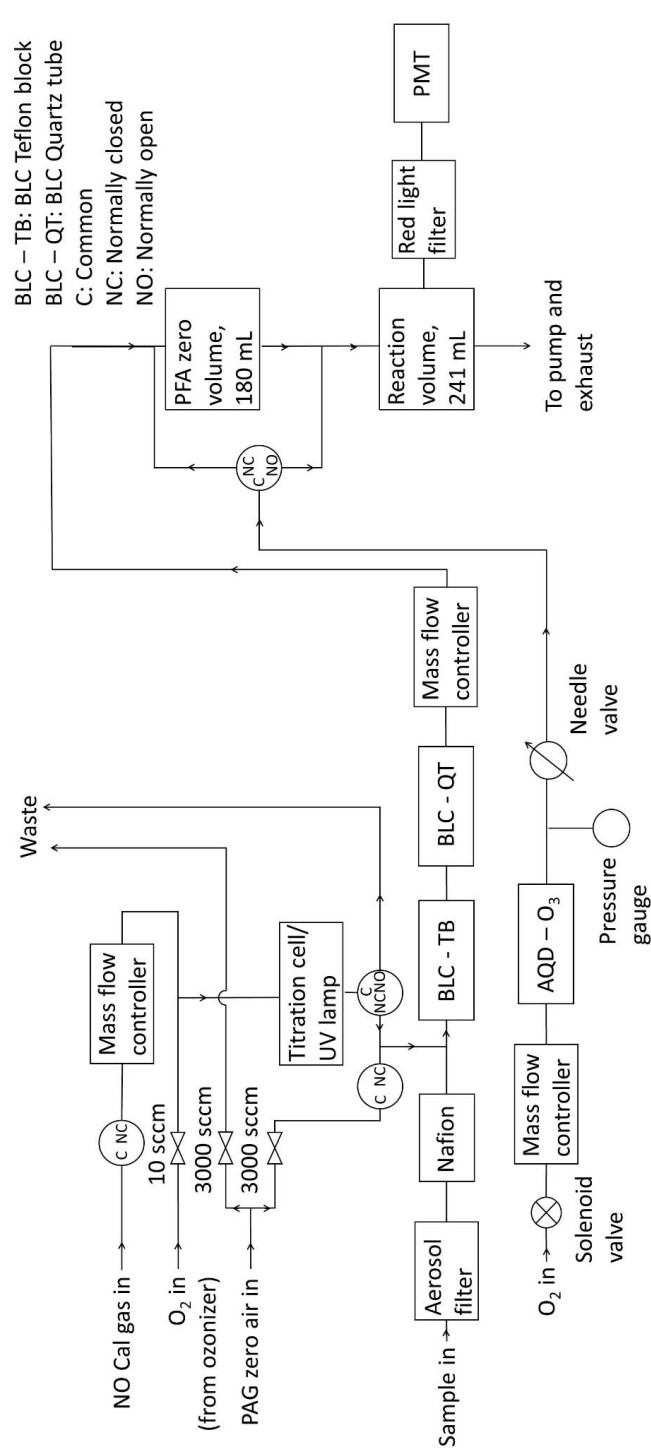

Figure 2: Flow diagram of the NOx instrument at the CVAO.





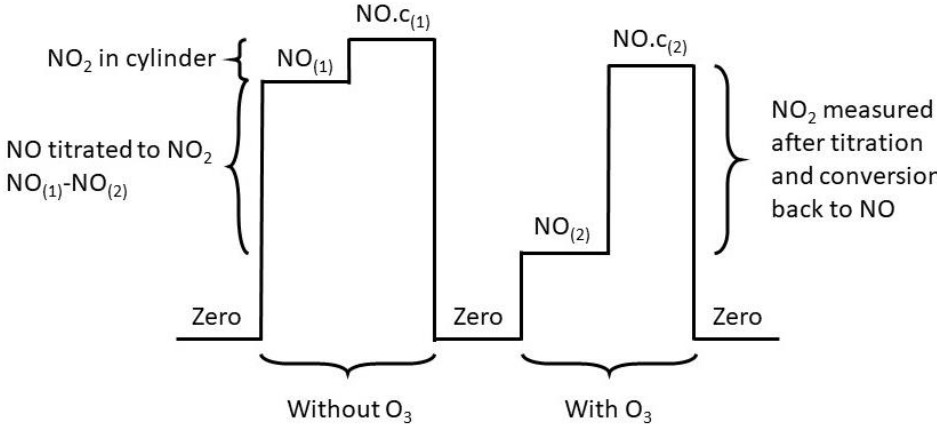

Figure 3: A theoretical calibration cycle. "NO" is the measurement of only NO i.e. when the converters are off, NO.c is when one of the converters are on therefore the measurement is NO + $NO_2$ and (1) and (2) represent untitrated and titrated NO, respectively.

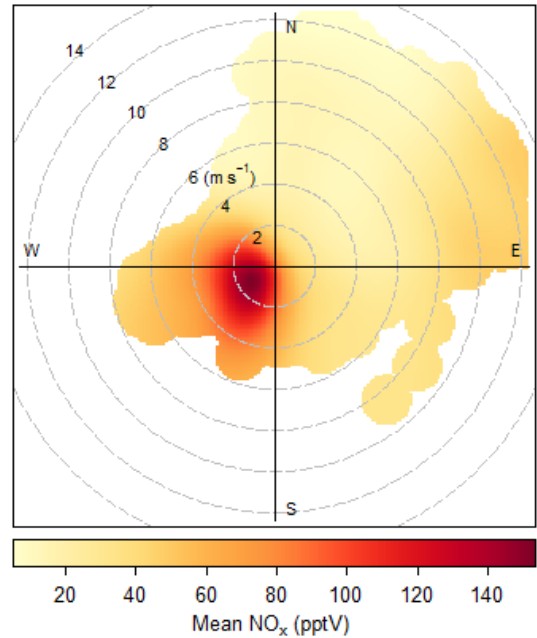

Figure 4: Total $NO_x$ from June 2017 to August 2019 plotted as a function of wind speed and direction.

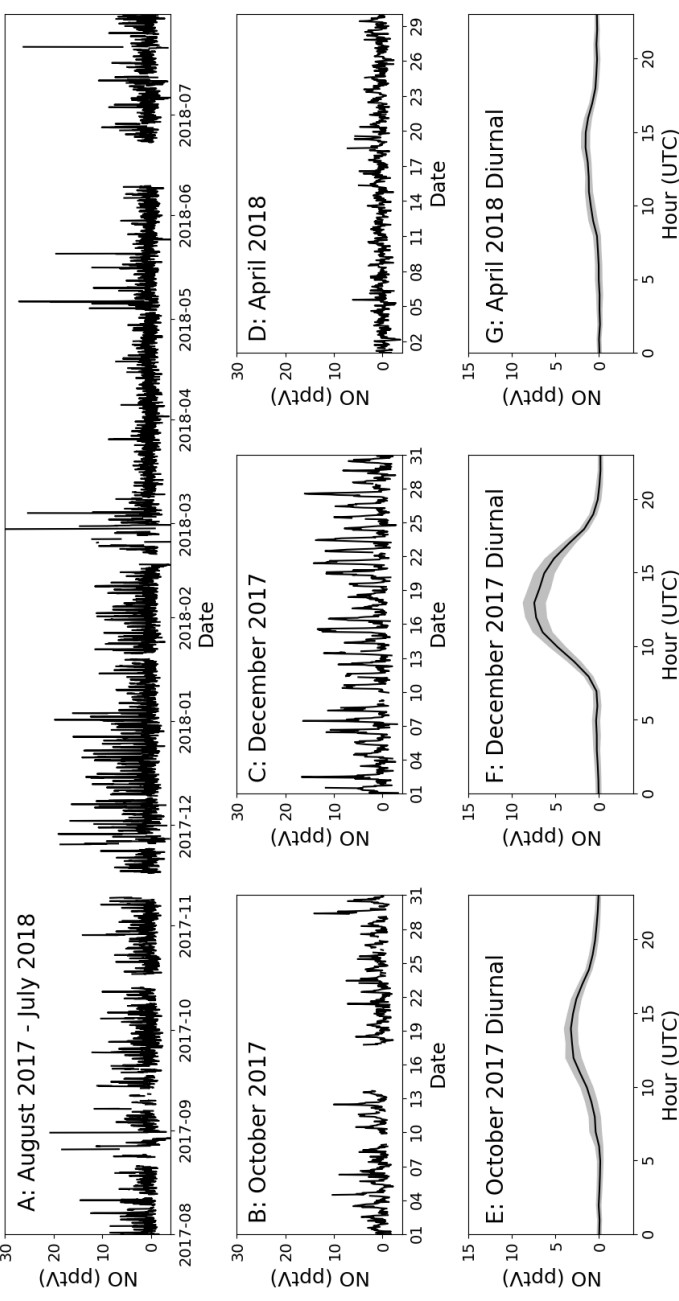

Figure 5: Panel A show the time series for filtered $O_3$ corrected NO from August 1st 2017 to July 31st 2018. Panel B, C and D zoom in on October 2017, December 2017 and April 2018, respectively. Panel E, F and G show the average diurnal of NO for October 2017, December 2017 and April 2018, respectively, with the coloured areas being ±2 standard errors. If there are less than 15 measurements available for the hour, it is not included in the diurnal.



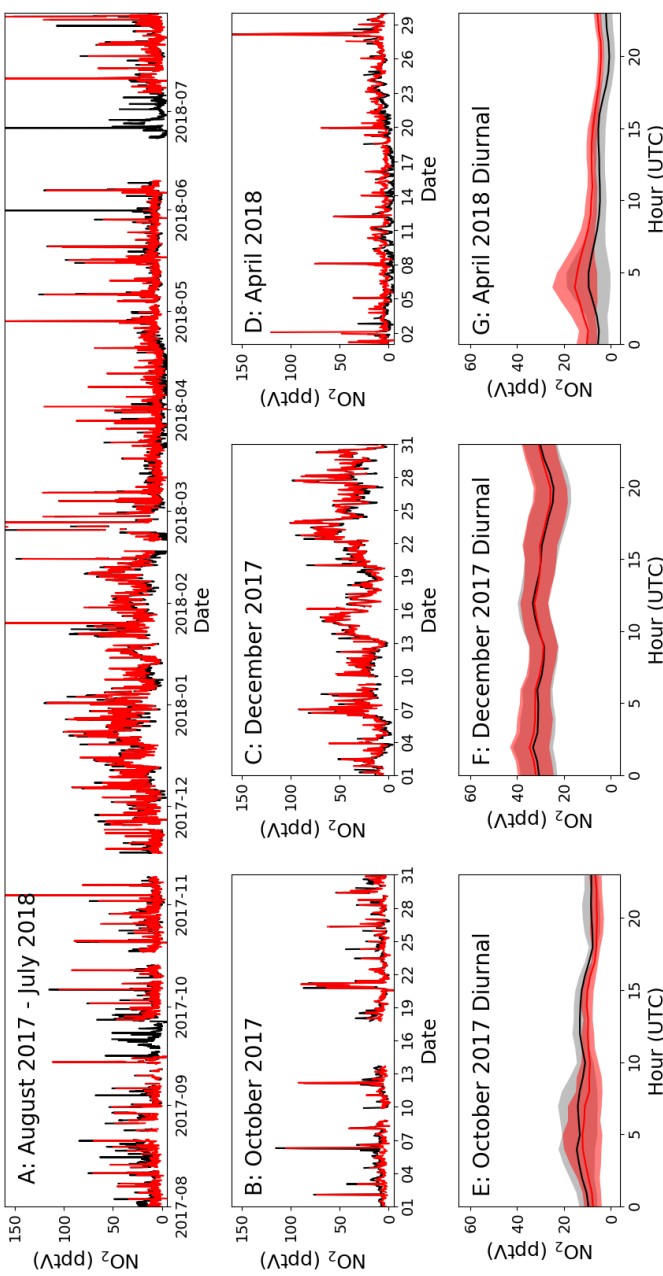

Figure 6: Panel A show the time series of filtered $O_3$ corrected $NO_2$ from August $1^{st}$ 2017 to July $31^{st}$ 2018 for the BLC (black) and PLC (red). Panel B, C and D zoom in on October 2017, December 2017 and April 2018, respectively, with the red line being the PLC and the black being the BLC. Panel E, F and G show the average diurnal of $NO_2$ for October 2017, December 2017 and April 2018, respectively, with the red line being the PLC and the black being the BLC and the coloured areas being ±2 standard errors. If there are less than 15 measurements available for the hour, it is not included in the diurnal.



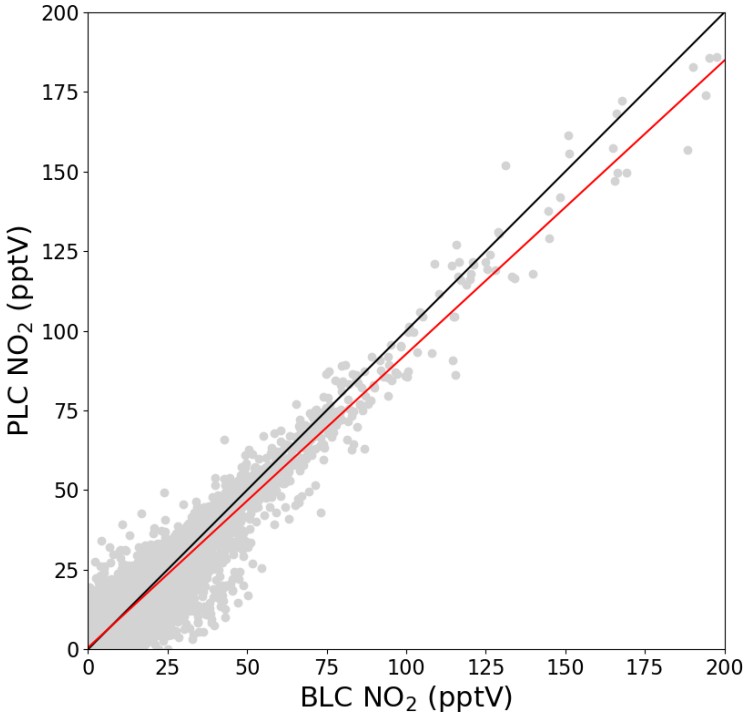

Figure 7: The PLC NO₂ concentration is plotted against the BLC NO₂ concentration. The black lines show the 1-to-1 relationship. The red line is the linear regression of the hourly data with uncertainties in both the x and y.

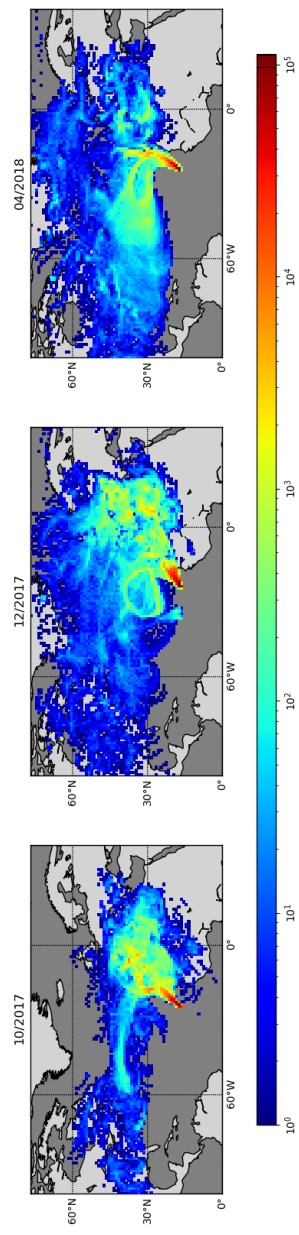

Figure 8: Back trajectories estimated for October 2017, December 2017, and April 2018.
FLEXPART version 10.4 is used in backwards mode, driven by pressure level data from Global
Forecast System (GFS) reanalyses at 0.5°×0.5° resolution (Pisso et al., 2019; Stohl et al., 1998).
10-day back-trajectory simulations are initialised every 6 hours, releasing 1000 particles from
the CVAO site.



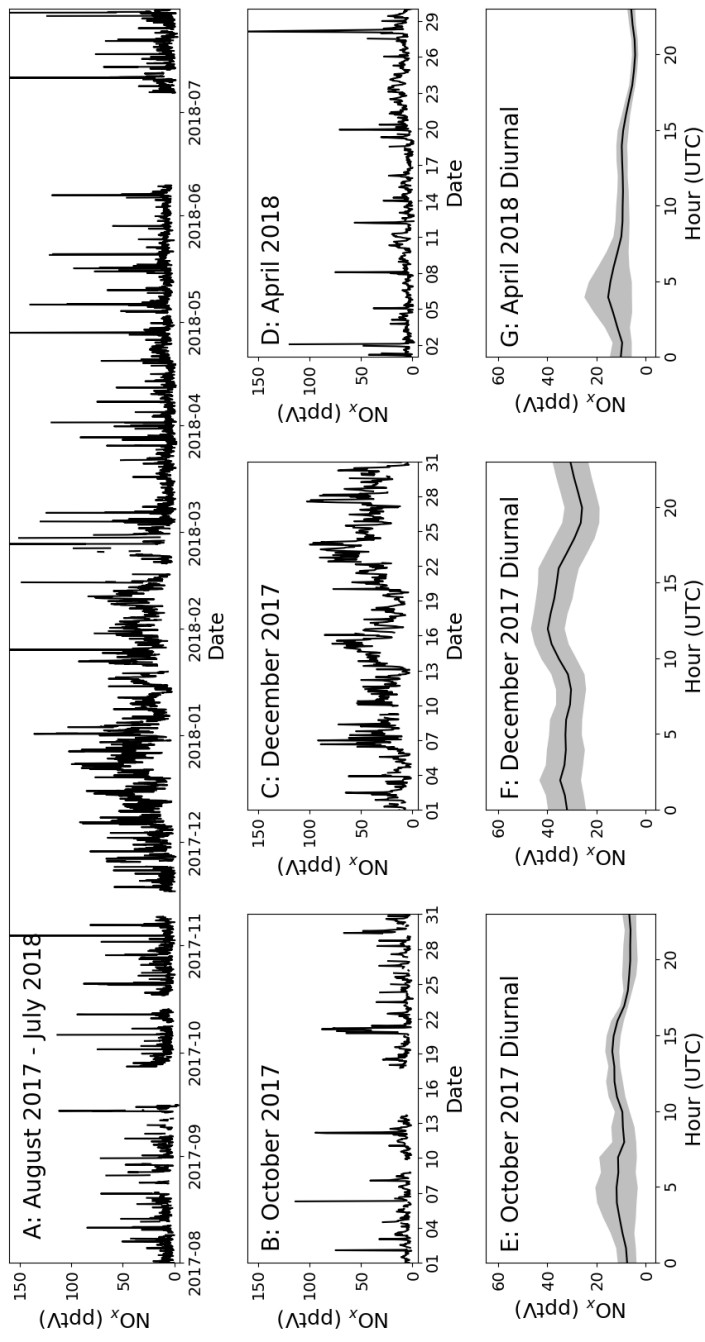

Figure 9: Panel A show the time series for total $NO_x$ (NO + $NO_2$ PLC) from August 1st 2017 to July 31st 2018. Panel B, C and D zoom in on October 2017, December 2017 and April 2018, respectively. Panel E, F and G show the average diurnal of $NO_x$ for October 2017, December 2017 and April 2018, respectively, with the coloured areas being ±2 standard errors. If there are less than 15 measurements available for the hour, it is not included in the diurnal.



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
