# Peer review of "Long-term NOx measurements in the remote marine tropical troposphere"

_Atmospheric Measurement Techniques, 2020_

## Referee Comment (RC1) · Anonymous Referee #1 · 8 Jan 2021

This paper reports on NOx measurements at the remote marine site of Cape Verde. The instrumental setup, the data processing, the correction of interferences, the analysis of artefacts and the uncertainty analysis is comprehensively described. Two converters, a commercially available blue light converter and a self-built photolytical converter were operated for two years in parallel and periodically checked for potential artefacts. A major finding of this paper is that a photolytical converter of the type presented here should preferably be used at remote sites due to its smaller artefacts compared to the commercially available BLC.

NOx measurements at remote marine sites are very challenging and only view datasets have been published so far. With its clear and concise structure this paper

could help to improve NOx measurements at remote sites and foster the harmonization of NOx measurements. Therefore, I would recommend that this paper be accepted for publication after considering few comments.

The determination of the NO artefact is challenging when measuring at remote sites. In this paper two methods are compared, using nocturnal NO data and zero air. Both methods agree well during most of the time, however at some instances they differ by more then 20 pptV which is higher than the maximum ambient values. I would also agree with the authors that using the nocturnal NO mixing ratios for artefact correction should be the preferred method. However, even at remote sites NO can be advected from nearby sources at low windspeeds or varying ozone concentrations. Has it been checked by auxiliary measurements that the deviation from NO artefact determinations are not caused by nearby contaminations?

One of the major results of this paper is from the concurrent operation of the two converters with the same CLD. The data of both converters agree well, however there is still a slope of 0.92 in the regression plot. This difference could be significant considering that measurements are done using the same calibration and the same detector. As the artefact the of the BLC is done by subtracting the measured $NO_2$ artefact, a wrong correction would only the zero and not the slope. Is it possible that there is an interference in the BLC observed at high $NO_2$ levels?

The BLC data showed an $NO_2$ artefact which was corrected by measuring PAG zero air. This variable artefact caused an offset in the PLC and BLC datasets of up to 10 pptV. Unfortunately, the plots of the PAG zero air measurements are not included in the paper or in the supplement. However, when looking at the $NO_2$ measurements from the NO cylinder in Figure S3, the difference between the converters is always less than 10 pptV. Can these measurements be used for the $NO_2$ offset correction?

Technical corrections:

Figure 2: In the flow diagram the photolytical converters are labelled as BLC-TB and BLC-QT while in the text they are reffered to as BLC and PLC. I would recommend using one designation only.

Line 74: MPAN and PAN belong to the group of acyl peroxy nitrates (APN). I would suggest rephrasing the sentence, e.g. reactive nitrogen species (NOz) such as peroxyacetyl nitrate (PAN), peroxymethacryloyl nitrate (MPAN) and other acyl peroxy nitrates (APN)

Line 75: $HNO_3$, $HO_2NO_2$, and HONO

Line 103: time zone of Cabo Verde is UTC-1

Line 310: converter

Line 367: mixing ratio

Line 331: over time, and

Line 471: caused by data of one morning

---

## Referee Comment (RC2) · Anonymous Referee #2 · 18 Jan 2021

Nitrogen oxides have an important role in the atmosphere regulating the key atmospheric oxidants O3 and OH. There are few long-term measurements of nitrogen oxides in the background atmosphere. This is one of a few papers that combine a detailed description of a measurement technique with multi-year observations of nitrogen oxides in the background atmosphere and it presents unique observations. After some issues are addressed, the paper merits publication in Atmospheric Measurement Techniques.

The aspect of the paper that requires additional detail, in the reviewers opinion, is the need to explicitly address the effect of humidity on the observations. The authors note that humidity affects the quenching of excited nitrogen dioxide and hence the instrument calibration. They do not explicitly note that humidity affects the background (zero) signal in these measurements via wall reactions associated with ozone that cause light

emissions in the reaction cell. This signal decreases at higher humidity. The point is that without a robust examination of the role of humidity, the uncertainty of the observations is most probably underestimated.

Lines 163-166 describes gas phase reactions and reactions in the photolytic convertor but not the ozone-surface reactions in the reaction chamber that makes up the majority of the zero minus thermionic noise signal. Prior studies have identified the role of water vapor in modifying the rates these surface reactions leading to the NO detection artifact. This needs correction. Line 392-393 While addressing humidity, the comment does not deal with it in any depth.

The authors dry their sample air with a Nafion dryer and use dry zero air for diluting their calibration gas. Two questions:

1. Is the absolute humidity in the reaction cell the same in calibration and measurement modes?

2. If not, what effect does this have on the calibration?

With regard to ambient measurements:

1. what variation in efficiency of water removal occurs on short and long time scales with the Nafion dryer in the course of ambient monitoring?

2. What effect will the associated variations in humidity in the reaction cell have on ambient measurements?

3. Are there tests of the NO and NO2 transmission of the Nafion dryer?

Minor Issues 1. The paper, Berkes et al. (2018) AMT covers related material from a different perspective. Including discussion of the Berkes paper would strengthen the current paper.

2. Abstract line 19. It would be appropriate to introduce NO detection before NO2 conversion.

3. Units. From line 57 onwards. pptV is not a SI unit. I suggest the authors consider including text something like the following: The appropriate SI unit is mole fraction, picomole/mole. It is assumed that under tropospheric conditions at the low mole fractions discussed, that NO and NO2 behave as ideal gases and therefore mole fraction is equivalent to volumetric mixing ratio. Volumetric mixing ratio is commonly used in the literature and the appropriate range here is represented by ppt, 10-12 mixing ratio by volume and ppt is used here. (Use of pptV is generally discouraged, see IUPAC review of units in atmospheric chemistry.)

4. Given the magnitude of the uncertainties and sd's recorded throughout the paper, the data in ppt could be rounded to one decimal place (0.1 ppt).

5. Line 126-130. Is the flow in the inlet laminar or turbulent? What is the transmission efficiency of the inlet line plus filter for NO and NO2?

6. Line 280-300. The discussion lacks mention of the role of humidity in artifacts. This needs additional discussion as identified above.

7. Line 281-283 The first sentence is unclear and requires revision.

8. Line 293 the word ordinarily may not be the best choice for this description.

9. Line 330-339 Were the convertors cleaned during the course of the observations? What was the cleaning procedure? What was the effect of cleaning on the conversion efficiency and the NO2 artifact?

10. Line 344 The two "believed" are really "assumed"? The authors should state the consequences for data interpretation if the assumptions are wrong as well as when they are correct. This section could be clearer if expressed as a set of simple equations.

11. Line 379 Wind direction does not appear to be correctly specified.

12. Line 507 A more general overview sentence would make a better introduction to the Conclusion section.

[Figure]

13. Table 1, the reviewer finds the caption, headers and footnote confusing. Please clarify.

14. Figure 7, please clarify: is the fitted red line a weighted least square fit, a reduced major axis regression or something else?

Supplementary Information

1. Line 48-89. The use of the term photo-stationary state appears inconsistent with the definition of photo-stationary state by IUPAC (please modify) see https://goldbook.iupac.org/terms/view/P04654 A steady state reached by a reacting chemical system when light has been absorbed by at least one of the components. At this state the rates of formation and disappearance are equal for each of the transient molecular entities formed.

2. Line numbers cease at line 121. For subsequent text check spelling, define relative uncertainty and indicate that relative uncertainty is being used in the evaluations.

3. Figure S2 Please include dimensions.

References The following references need completion/correction.

Bell, S.

Buhr, M.P.

Drummond, J.W.

Gilge, S.

The following is probably the correct reference and web location for the reference Galbally (2019) cited in the paper.

Galbally, I.E. (2020). Nitrogen Oxides (NO, NO2, NOy) measurements at Cape Grim: A technical manual. In 'Baseline Atmospheric Program (Australia): Technical Series'. (eds. S.J. Cleland, N. Derek and P.B. Krummel). Bureau of Meteorology and CSIRO

Oceans and Atmosphere: Melbourne Australia, v, 111p. https://doi.org/10.25919/dt6y-3q53

The following reference is relevant and should be included in the background and discussion:

Florian Berkes, Norbert Houben, Ulrich Bundke, Harald Franke, Hans-Werner Pätz, Franz Rohrer, Andreas Wahner, and Andreas Petzold. The IAGOS NOx instrument – design, operation and first results from deployment aboard passenger aircraft. Atmos. Meas. Tech., 11, 3737–3757, https://doi.org/10.5194/amt-11-3737-2018, 2018
* * *

---

## Author Comment (AC1) · 2 Mar 2021

The authors would like to thank both reviewers for their thorough reviews and good questions. All comments and questions have been responded to point by point in the attached pdf.

Please also note the supplement to this comment:
https://amt.copernicus.org/preprints/amt-2020-469/amt-2020-469-AC1-supplement.pdf

---

## Author Response (AR1)

The authors would like to thank both reviewers for their thorough reviews and good questions. All comments and questions have been responded to point by point below in red.

Anonymous Referee #1

This paper reports on NOx measurements at the remote marine site of Cape Verde. The instrumental setup, the data processing, the correction of interferences, the analysis of artefacts and the uncertainty analysis is comprehensively described. Two converters, a commercially available blue light converter and a self-built photolytical converter were operated for two years in parallel and periodically checked for potential artefacts. A major finding of this paper is that a photolytical converter of the type presented here should preferably be used at remote sites due to its smaller artefacts compared to the commercially available BLC.

NOx measurements at remote marine sites are very challenging and only view datasets have been published so far. With its clear and concise structure this paper could help to improve NOx measurements at remote sites and foster the harmonization of NOx measurements. Therefore, I would recommend that this paper be accepted for publication after considering few comments.

The determination of the NO artefact is challenging when measuring at remote sites. In this paper two methods are compared, using nocturnal NO data and zero air. Both methods agree well during most of the time, however at some instances they differ by more then 20 pptV which is higher than the maximum ambient values. I would also agree with the authors that using the nocturnal NO mixing ratios for artefact correction should be the preferred method. However, even at remote sites NO can be advected from nearby sources at low windspeeds or varying ozone concentrations. Has it been checked by auxiliary measurements that the deviation from NO artefact determinations are not caused by nearby contaminations?

The prevailing (>95%) winds at the CVAO are straight off the Atlantic and are extremely clean. Measurements in winds which come across the island before reaching the observatory and/or are associated with low wind speed are not used in the analyses. Additionally, if the nocturnal NO mixing ratio differs by more than 6.5 ppt between subsequent nights, then the data is not used, further avoiding use of data which could be influenced by any local pollution source. The PAG measurements can be observed to be significantly higher than the nocturnal measurements in 2019, which can be explained by issues with the compressor supplying the

PAG. In 2016 the PAG measurements can be observed to be significantly lower than the nocturnal measurements, which could be due to contamination, however, the $NO_2$ measurements from the PAG are not significantly different during this period compared to the rest of the year (a figure of the PAG $NO_2$ measurements has been added to the supplementary (Figure S8)), suggesting an interference in the background measurement during that period. Text has been added to figure S7 to explain the discrepancies.

One of the major results of this paper is from the concurrent operation of the two converters with the same CLD. The data of both converters agree well, however there is still a slope of 0.92 in the regression plot. This difference could be significant considering that measurements are done using the same calibration and the same detector. As the artefact the of the BLC is done by subtracting the measured NO2 artefact, a wrong correction would only the zero and not the slope. Is it possible that there is an interference in the BLC observed at high NO2 levels?

To further investigate the correlation of the two measurements, we have chosen to plot the BLC measurements as a function of the PLC measurements instead of the other way around, which was originally in the paper. This is done to perform a linear least squares regression besides the orthogonal distance regression and as no artefact correction is applied to the PLC measurements, it has a lower uncertainty and is therefore more appropriate to use as the "known" value in the linear least squares regression. The linear least squares regression gives a fit of $BLC = 0.99 \times PLC + 0.7$ ppt and an orthogonal distance regression where an uncertainty is given to both measurements gives $BLC = 1.08 \times PLC - 0.6$ ppt. The deviation in the slope from 1 in both regressions are consistent with the uncertainty in the measured $NO_2$ artefact which has been determined to be $7.2 \pm 7.2$ ppt. A new figure with both regression fits have replaced the original figure and text has been added to explain the different fits.

The BLC data showed an NO2 artefact which was corrected by measuring PAG zero air. This variable artefact caused an offset in the PLC and BLC datasets of up to 10 pptV. Unfortunately, the plots of the PAG zero air measurements are not included in the paper or in the supplement. However, when looking at the NO2 measurements from the NO cylinder in Figure S3, the difference between the converters is always less than 10 pptV. Can these measurements be used for the NO2 offset correction?

There is an artefact in the BLC measurements, which is corrected using the PAG zero measurements, however, the PAG air is not completely void of $NO_2$. The PLC measurements are assumed to not have an artefact and therefore the PAG PLC measurements are used to correct the PAG BLC measurements. The PAG PLC measurements are the ones varying by 0-10 ppt. A figure of the PAG zero $NO_2$ measurements have now been included in the supplementary information (Figure S8) to show the variability. The $NO_2$ measurements in the NO cylinder are given as a percentage and can therefore not be compared to the up to 10 ppt of $NO_2$ in the PAG air.

Technical corrections:

Figure 2: In the flow diagram the photolytical converters are labelled as BLC-TB and BLC-QT while in the text they are reffered to as BLC and PLC. I would recommend using one designation only.

Thank you for noticing this discrepancy. It has now been corrected.

Line 74: MPAN and PAN belong to the group of acyl peroxy nitrates (APN). I would suggest rephrasing the sentence, e.g. reactive nitrogen species (NOz) such as peroxyacetyl nitrate (PAN), peroxymethacryloyl nitrate (MPAN) and other acyl peroxy nitrates (APN)

The sentence has now been rephrased.

Line 75: HNO3, HO2NO2, and HONO

HO2NO2 has been added to the sentence.

Line 103: time zone of Cabo Verde is UTC-1

Line 310: converter

Line 367: mixing ratio

Line 331: over time, and

Line 471: caused by data of one morning

All of the above mentioned technical corrections have been followed. Thank you to the reviewer for noticing them.

Anonymous Referee #2

Nitrogen oxides have an important role in the atmosphere regulating the key atmospheric oxidants O3 and OH. There are few long-term measurements of nitrogen oxides in the background atmosphere. This is one of a few papers that combine a detailed description of a measurement technique with multi-year observations of nitrogen oxides in the background atmosphere and it presents unique observations. After some issues are addressed, the paper merits publication in Atmospheric Measurement Techniques.

The aspect of the paper that requires additional detail, in the reviewers opinion, is the need to explicitly address the effect of humidity on the observations. The authors note that humidity affects the quenching of excited nitrogen dioxide and hence the instrument calibration. They do not explicitly note that humidity affects the background (zero) signal in these measurements via wall reactions associated with ozone that cause light emissions in the reaction cell. This signal decreases at higher humidity. The point is that without a robust examination of the role of humidity, the uncertainty of the observations is most probably underestimated.

While all of these comments are correct, the background signal is measured every 5 minutes, so any impacts of changing humidity on the background signal are taken into account. Additionally, the sensitivity of the instrument can be observed to be stable between measurements in figure S4. The authors appreciate the reviewer's comments however and have added text to explain the role of humidity further.

Lines 163-166 describes gas phase reactions and reactions in the photolytic convertor but not the ozone-surface reactions in the reaction chamber that makes up the majority of the zero minus thermionic noise signal. Prior studies have identified the role of water vapor in modifying the rates these surface reactions leading to the NO detection artifact. This needs correction.

Text has been added to include a description of ozone-surface reactions and the role of water vapour in the NO artefact.

Line 392-393 While addressing humidity, the comment does not deal with it in any depth.

Additional text has been added about humidity in other places in the paper, however, this specific place discusses corrections performed on the data and since no humidity correction is performed, the authors don't feel there is a need to add more information here.

The authors dry their sample air with a Nafion dryer and use dry zero air for diluting their calibration gas. Two questions:

1. Is the absolute humidity in the reaction cell the same in calibration and measurement modes?

2. If not, what effect does this have on the calibration?

While the sample air is dried with a Nafion dryer, the calibration gas is not diluted with zero air. The calibration is done by standard addition to the sample air, so the humidity is the same during measurement and calibration mode, with the exception of when measuring the $NO_2$ artefact, which is measured in dry zero air.

With regard to ambient measurements:

1. what variation in efficiency of water removal occurs on short and long time scales with the Nafion dryer in the course of ambient monitoring?

2. What effect will the associated variations in humidity in the reaction cell have on ambient measurements?

There is no measurement of the humidity before and after the Nafion dryer, but the variation in the humidity after the Nafion dryer is expected to be very low, which is supported by the sensitivity measurements being stable within 5%.

3. Are there tests of the NO and NO2 transmission of the Nafion dryer?

No laboratory tests have been made to evaluate the NO and $NO_2$ transmission of the Nafion dryer, however, when measuring without the Nafion dryer for a week no significant changes were observed in the measurements from when measuring with the Nafion dryer.

Minor Issues

1. The paper, Berkes et al. (2018) AMT covers related material from a different perspective. Including discussion of the Berkes paper would strengthen the current paper.

The reference has been added in the introduction, when discussing the need for NOx measurements in remote areas.

2. Abstract line 19. It would be appropriate to introduce NO detection before NO2 conversion.

The major focus of the paper is the comparison of two $NO_2$ converters, therefore, the authors would like to keep the order of the abstract.

3. Units. From line 57 onwards. pptV is not a SI unit. I suggest the authors consider including text something like the following: The appropriate SI unit is mole fraction, picomole/mole. It is assumed that under tropospheric conditions at the low mole fractions discussed, that NO and NO2 behave as ideal gases and therefore mole fraction is equivalent to volumetric mixing ratio. Volumetric mixing ratio is commonly used in the literature and the appropriate range here is represented by ppt, 10-12 mixing ratio by volume and ppt is used here. (Use of pptV is generally discouraged, see IUPAC review of units in atmospheric chemistry.)

Thank you. Throughout the paper, and in the figures pptV has been changed to ppt and the suggested text has been included.

4. Given the magnitude of the uncertainties and sd's recorded throughout the paper, the data in ppt could be rounded to one decimal place (0.1 ppt).

Agreed - the measurements have been rounded to one decimal place.

5. Line 126-130. Is the flow in the inlet laminar or turbulent? What is the transmission efficiency of the inlet line plus filter for NO and NO2?

With a Reynolds number >10.000, the flow in the inlet is turbulent. A 40 mm glass manifold is used to avoid loss of gases on the manifold and PTFE filters are commonly used at the inlet of NOx instruments to remove particles, however, the transmission efficiency has not been measured.

6. Line 280-300. The discussion lacks mention of the role of humidity in artifacts. This needs additional discussion as identified above.

Text has been added to this section to make it clear that the risk of the humidity being different could cause the artefacts to be either under- or overestimated.

7. Line 281-283 The first sentence is unclear and requires revision.

The authors cannot see an issue with the sentence.

8. Line 293 the word ordinarily may not be the best choice for this description.

"ordinarily" has been changed to "generally".

9. Line 330-339 Were the convertors cleaned during the course of the observations? What was the cleaning procedure? What was the effect of cleaning on the conversion efficiency and the NO2 artifact?

The converters have not been cleaned since the conversion efficiency and NO2 artefact has remained stable throughout the measurements.

10. Line 344 The two "believed" are really "assumed"? The authors should state the consequences for data interpretation if the assumptions are wrong as well as when they are correct. This section could be clearer if expressed as a set of simple equations.

"believed" has been changed to "assumed". If the assumptions are wrong and the PLC do have a significant artefact, then both $NO_2$ measurements will be overestimated. The PLC measurements would include an artefact and the BLC artefact would be underestimated if the PLC PAG measurements are a combination of artefact and $NO_2$, which would mean a higher BLC artefact should be subtracted. During the summer, the $NO_2$ measurements go as low as ~5 ppt with the current assumptions, it is therefore, believed to be a valid assumption.

11. Line 379 Wind direction does not appear to be correctly specified.

The wind direction has been specified.

12. Line 507 A more general overview sentence would make a better introduction to the Conclusion section.

A sentence has been added to the conclusion.

13. Table 1, the reviewer finds the caption, headers and footnote confusing. Please clarify.

The captions, headers, and footnotes have been changed to be more clear.

14. Figure 7, please clarify: is the fitted red line a weighted least square fit, a reduced major axis regression or something else?

It was an orthogonal distance regression fit, which was mentioned in the text, but not in the figure text, where it has now been added.

Supplementary Information

1. Line 48-89. The use of the term photo-stationary state appears inconsistent with the definition of photo-stationary state by IUPAC (please modify) see https://goldbook.iupac.org/terms/view/P04654 A steady state reached by a reacting chemical system when light has been absorbed by at least one of the components. At this state the rates of formation and disappearance are equal for each of the transient molecular entities formed.

The photo-stationary state described in the O3 correction is the point in either of the photolytic converters, where the photolysis rate of NO2 is equal to the reaction of NO+O3, therefore, the term will continue to be used.

2. Line numbers cease at line 121. For subsequent text check spelling, define relative uncertainty and indicate that relative uncertainty is being used in the evaluations.

Line numbers have been added and the uncertainty analysis has been checked for spelling. The relative uncertainty is only used in the supplementary and the actual uncertainty in ppt is used inside the paper.

3. Figure S2 Please include dimensions.

Dimensions have been added to the figure text.

References

The following references need completion/correction.

Bell, S.

Buhr, M.P.

Drummond, J.W.

Gilge, S.

The following is probably the correct reference and web location for the reference Galbally (2019) cited in the paper. Galbally, I.E. (2020). Nitrogen Oxides (NO, NO2, NOy) measurements at Cape Grim: A technical manual. In 'Baseline Atmospheric Program (Australia): Technical Series'. (eds. S.J. Cleland, N. Derek and P.B. Krummel). Bureau of Meteorology and CSIRO Oceans and Atmosphere: Melbourne Australia, v, 111p. https://doi.org/10.25919/dt6y3q53

Thank you for a very thorough review, the references have been corrected accordingly.

The following reference is relevant and should be included in the background and discussion: Florian Berkes, Norbert Houben, Ulrich Bundke, Harald Franke, Hans-Werner Pätz, Franz Rohrer, Andreas Wahner, and Andreas Petzold. The IAGOS NOx instrument – design, operation and first results from deployment aboard passenger aircraft. Atmos. Meas. Tech., 11, 3737–3757, https://doi.org/10.5194/amt-11-3737-2018, 2018

The reference has been added.